# Whole Corneal Descemetocele

**DOI:** 10.3390/medicina59101780

**Published:** 2023-10-06

**Authors:** Mao Kusano, Yasser Helmy Mohamed, Masafumi Uematsu, Daisuke Inoue, Kohei Harada, Diya Tang, Takashi Kitaoka

**Affiliations:** Department of Ophthalmology and Visual Sciences, Graduate School of Biomedical Sciences, Nagasaki University, Nagasaki 852-8501, Japan; maok@nagasaki-u.ac.jp (M.K.); uematsu1124@outlook.jp (M.U.); d.i.private.3@gmail.com (D.I.); nagasakidejima@yahoo.co.jp (K.H.); bb55319802@ms.nagasaki-u.ac.jp (D.T.); tkitaoka@nagasaki-u.ac.jp (T.K.)

**Keywords:** descemetocele, microbial keratitis, penetrating keratoplasty (PRP)

## Abstract

*Background and Objectives*: To report a case of microbial keratitis complicated by severe corneal melting and whole corneal descemetocele. *Methods*: A 72-year-old male farmer presented with a right corneal ulcer involving nearly the entire cornea, which was almost completely melted down with the remaining Descemet’s membrane (DM). The pupil area was filled with melted necrotic material, with the intraocular lens partially protruding from the pupil and indenting the DM. Corneal optical coherence tomography (OCT) examination revealed a corneal thickness of 37 µm that was attached to its back surface, with the iris and a part of the intraocular lens (IOL) protruding through the pupil. The patient was hospitalized and treated with local and systemic antibiotics until control of the inflammation was achieved. Corneoscleral transplantation plus excision/transplantation of the corneal limbus were performed, and the entire corneal limbus was lamellarly incised. After completely suturing all around the transplanted corneoscleral graft, the anterior chamber was formed. Postoperative treatment included local antibiotics, anti-inflammatory drugs, and cycloplegic drops. *Results*: There was no recurrence of infection, and the corneal epithelium gradually regenerated and covered the whole graft. Visual acuity was light perception at 6 months after the surgery. The patient was satisfied that the globe was preserved and did not wish to undergo any further treatment. *Conclusions*: Corneoscleral transplantation is preferred for the treatment of large-sized descemetoceles with active microbial keratitis and extensive infiltrates, especially in cases where the whole cornea has transformed into a large cyst.

## 1. Introduction

A corneal descemetocele, which is the anterior herniation of an intact Descemet’s membrane (DM) through an overlying stromal defect, is a rare but serious outcome of progressive corneal ulceration and requires urgent intervention due to an imminent risk of perforation [1,2,3,4,5]. As there is a lack of adequate tensile strength, the DM herniates anteriorly with an appearance similar to a cyst that protrudes through the overlying corneal stromal defect. This is referred to as a descemetocele [6].

Various ocular and systemic abnormalities that can lead to the formation of descemetocele include microbial keratitis, neurotrophic keratopathy, dry eye disorders, and corneal inflammation associated with immune-mediated disorders [6]. Corneal transplantation is mandatory if the infectious corneal ulcer results in severe corneal tissue melting. Furthermore, when there is an infiltrating corneal lesion that infiltrates deeper into the stromal thickness, prompt intervention is required to restore the ocular structural integrity [5].

An infectious corneal ulcer is considered an eye emergency if it progresses to corneal descemetocele and perforation. Extensive medical treatment with suitable antimicrobial eye drops is mandatory. Surgical interventions, such as the corneal patch with tissue adhesives, amniotic membrane grafts, and conjunctival flaps, are indispensable to restore ocular anatomy and decrease side effects [6,7]. Keratoplasty techniques are alternative procedures that are utilized for the purpose of preserving eye anatomy and include lamellar keratoplasty and penetrating keratoplasty (PKP) [8]. Surgery aims to maintain or restore corneal integrity, eradicate infection, and stabilize or improve visual acuity.

To the best of our knowledge, this was the largest descemetocele ever to be reported in the literature, with the authors also describing a rare case of severe corneal melting. In this case, the entire cornea was transferred to descemetocele, which we refer to as a whole corneal descemetocele.

## 2. Case Report

A 72-year-old male patient presented to his local ophthalmologist with pain and loss of vision in his right eye. The patient was a farmer and had previously undergone bilateral cataract surgery seven years earlier. There was no history of any recent eye trauma. He had a history of diabetes mellitus on medication, and family history was irrelevant. Examination revealed a large, centrally infected corneal ulcer in the patient’s right eye. The ulcer was accompanied by a grayish-white infiltration that involved the central cornea and extended to the lower periphery (Figure 1A). The best-corrected visual acuity was counting fingers at 10 cm in the right eye and at 1.5 cm in the left eye. The intraocular pressure (IOP) was normal in both eyes. The patient was prescribed 1.5% levofloxacin, 0.1% cefmenoxime hydrochloride, and dibekacin sulfate eye drops every two hours. In addition to the use of ofloxacin ointment at bedtime, he was also administered oral levofloxacin 500 mg/day. There was no improvement noted at two weeks, with deterioration of both the corneal ulcer and infiltration (Figure 1B). Therefore, the ofloxacin ointment was replaced with pimaricin ointment, while the oral levofloxacin was changed to fluconazole 200 mg/day. Three weeks after the initial visit, the patient was referred to our clinic due to further deterioration of the corneal ulcer, melting of the corneal stroma, and a lost depth of the anterior chamber (Figure 1C).

The initial examination at the Ophthalmology Department of Nagasaki University Hospital revealed the presence of a right corneal ulcer that involved nearly the entire cornea and that had almost completely melted down with the remaining DM (Figure 2A,B). The pupil area was filled with melted necrotic material, and the intraocular lens was partially protruding from the pupil and was indenting the DM (Figure 2B). The iris was shown to be anteriorly bowed, with 360-degree rubeosis iridis along with lost anterior chamber depth (Figure 2B). In addition, there was also conjunctival and scleral hyperemia around the globe, and the sclera was slightly edematous. The fundus could not be seen, and the visual acuity was hand movement with soft IOP on palpation.

Corneal optical coherence tomography (OCT) examination revealed 37 µm of corneal thickness attached to its back surface with the iris, and part of the IOL was protruding through the pupil and indenting the DM (Figure 2C,D).

A conjunctival swab was taken from the patient for culture and sensitivity in a trial to identify the causative organism.

The patient was hospitalized and treated with additional topical 1.0% vancomycin 6 times/day, along with a 2 g infusion of cefazolin sodium twice daily. Gradual corneal epithelial regeneration was observed on the remaining DM, ranging from the peripheral to the central cornea (Figure 3A,B). In addition, there was a gradual improvement in the inflammation and hyperemia of the conjunctiva and sclera. On the seventh day of hospitalization, a fundus examination was possible, and the macula and optic nerve appeared normal. Although mild choroidal detachment was observed in the periphery of the fundus, there was no obvious inflammatory spillover to the posterior segment of the eye. Smears and cultures performed at the time of the initial examination were negative for bacteria and/or fungi.

After the corneal inflammation had settled down within two weeks of hospitalization and intensive treatment, corneoscleral transplantation and excision/transplantation of the corneal limbus were performed. The entire corneal limbus was lamellarly incised, and an abscess near the pupil was excised and sent for PCR. The preserved corneoscleral graft was sutured all around (Figure 3C), which allowed for the formation of the anterior chamber. The excised corneal limbal tissue was then once again transplanted around the donor cornea, with a continuous suture with the conjunctiva also performed.

Postoperative treatment included 1.5% levofloxacin, 1.0% vancomycin, and 0.1% betamethasone sodium phosphate eye drops six times daily. In addition, 1% atropine sulfate hydrate eye drops twice daily and oral levofloxacin and acetazolamide 500 mg/day were prescribed. The PCR result was negative for causative organisms. Postoperatively, the patient had anterior chamber coagulum and hyphema and vitreous hemorrhage, probably due to severe iris rubeosis. OCT revealed a formed anterior chamber with hyphema (Figure 3D). Fundus photography was not possible in this case. There was no recurrence of the infection, and the corneal epithelium gradually regenerated and covered the whole graft area. Visual acuity was light perception at 6 months after the surgery. The patient was satisfied that the globe had been preserved and did not wish to undergo any further treatment.

## 3. Discussion

A descemetocele is defined as the exposure or protrusion of the DM on the anterior surface of the cornea [1]. DM is an 8–10 µm thick, transparent, elastic, and acellular membrane that is secreted by endothelial cells [3]. It is relatively resistant to proteolysis and biomechanical stress and protects the endothelium from destructive stromal processes [9].

Descemetoceles are divided according to their location into central: within ≤5 mm from the corneal center, paracentral: 5–8 mm, and peripheral: ≥8 mm (also including the limbal area). The classification is also according to its size at the maximum dimension, with small: <3 mm, medium: 3–6 mm, and large: >6 mm [10]. Kim et al. reported a patient with corneal edema and an opacity with 7 × 7 mm sized epithelial defects. The patient had undergone previous PKP for pseudophakic bullous keratopathy three years prior to the epithelial defect [5]. In another recent prospective interventional study, Shanker et al. presented details on a descemetocele that was 7.1 mm and caused by a thermochemical (blast) injury [10]. Previous literature has reported that the maximum extent of a descemetocele does not exceed 7.1 mm [1,2,5,6,7,9,10]. Thus, to the best of our knowledge, this is the first report to describe the conversion of the entire cornea into the descemetocele, which we refer to as the whole corneal descemetocele.

Anterior segment optical coherence tomography (OCT) is a useful technique for imaging corneal architecture. In cases where clinical visualization is hindered by overlying debris and mucoid discharge, this modality might help in perceiving the true stromal thickness and demonstrating a bulging DM [4,6]. It may also help in differentiating descemetocele from other abnormalities. Follow-up scans can aid in its monitoring by demonstrating increased thickness upon healing [4,5].

In our case, corneal OCT examination revealed 37 um of corneal thickness attached to its back surface with the iris, and part of the IOL was protruding through the pupil and indenting the DM.

The DM may or may not have an anterior herniation and, unless perforated, maintains its intactness [11]. The overlying stroma is characteristically thin with compromised epithelial integrity [12,13]. The margins of the ulcer may be clean with smooth DM and absent cellular infiltration or edematous with severe stromal melting and an abscess, depending on the severity and type of underlying cause [14,15]. There can be associated purulent discharge, hyperemic conjunctiva, ciliary injection, limbal involvement, and/or rapid progression of corneal infiltrates in severe keratitis [16,17].

Corneal perforation or the development of a descemetocele from microbial keratitis may lead to severe vision loss and ocular morbidity due to secondary glaucoma, cataracts, and endophthalmitis, thereby potentially requiring evisceration of the eye [18]. The basic steps in the treatment of this condition include the management of the underlying disease, suppression of inflammation, and hastening of the recovery process. Moreover, an urgent and effective treatment needs to be applied to achieve anatomic ocular integrity in the descemetocele and corneal perforations [19]. After analyzing a manner of experimental keratitis in rabbits incited by various strains of *Pseudomonas*, Twinning et al. were able to postulate the role of proteases in the formation of a descemetocele [1]. These authors demonstrated that *Pseudomonas* strains (102, 115, 118) had a higher production of protease, the presence of added Ca++ and Mg++, and that there were fewer inflammatory cells required to induce massive destruction of the stromal matrix, rapid descemetocele formation, and perforation as compared to the other strains (107, 111, 113). As a result, this caused a slight, trivial degeneration of the stroma despite an extensive inflow of leucocytic cells [1]. They concluded that descemetocele genesis was directly correlated with alkaline protease, total protease, and elastase activity. Moreover, although leukocytic proteases are shared in corneal degeneration, their existence does not necessarily lead to the formation of descemetocele [6]. In this previous report, intensive local and systemic treatment was used to try and control the microbial keratitis and inflammation to suppress the proteolytic activity, thereby preventing more thinning of the cornea and perforation.

Treatment options include therapeutic contact lenses [18,20], tissue glues [18,21], conjunctival flaps [18,22], amniotic membrane transplantation (AMT) [23,24], keratoplasty (penetrating, lamellar), and patch grafts [6,7,8,9,10,18,19,25]. There are both advantages and disadvantages to each of these treatment options.

Keratoplasty for therapeutic/tectonic purposes includes keratoplasty (penetrating and lamellar) and patch grafts [6]. The application and prognosis depend on the localization, depth, and size of the descemetocele or the perforated area. The success rate of therapeutic/tectonic keratoplasty is dependent on the disease process, the appropriateness of the selected surgical procedure, postsurgical factors that affect ocular surface integrity, and the degree of inflammation control [18,26]. Therapeutic keratoplasties performed for infections have a more favorable prognosis than those performed for factors causing corneal melting [19]. Corneal transplantations offer the insight of prolonged control of the corneal disease, in contrast to most other treatments, which are only short-term solutions. For many years, PKP has been considered the ideal procedure for a corneal perforation or descemetocele [6,27].

Descemetocele is considered a rare complication, and despite the scanty peer-reviewed information comparing the effects of different management options, there has been ambiguity with regard to the management of descemetocele. In 1984, Arentsen et al. studied the causes, management, and results of patients with descemetocele [2]. However, they did not compare the various management options and neglected the effect of various parameters on surgical results. In 2018, Ozdemir et al. evaluated the effect of numerous variables such as age, gender, cause, and location of descemetocele, along with the presence and size of the perforation, on the surgical results of descemetoceles [19]. Nevertheless, their study was limited to only two surgical interventions, PKP and AMT, with their results proving the superiority of PKP over AMT.

The type of keratoplasty performed (PRP versus lamellar) has a significant bearing with regard to the final anatomical and functional results. Overall, we prefer keratoplasty (corneoscleral transplantation) for descemetoceles in cases found to have progressive infectious keratitis and extensive infiltrates to save the cornea from infectious material and allow significant penetration of antimicrobials [10].

In the present study, our only options were to control and suppress the inflammation along with the use of PRP as a corneal graft due to the size of the descemetocele that involved the whole cornea. Although the priority of the treatment is to achieve globe integrity, improvement of the visual prognosis by means of therapy is also important [19]. While therapeutic success has been defined as the ability to maintain ocular integrity at all follow-ups without any need for repeat tectonic procedures, the functional outcome was considered reasonable if the patient gained a visual acuity >3/60 in the operated eye [10].

In the present study, although the patient’s final visual acuity was perception of light, the patient was very satisfied due to the disappearance of the pain and inflammation, as well as the preservation of his globe anatomy. As a result, this patient requested that there be no further procedures.

In conclusion, PKP (corneoscleral transplantation) is preferred for large-sized descemetoceles with active microbial keratitis and extensive infiltrates, especially in cases where the whole cornea has transformed into a large cyst.

## Figures and Tables

**Figure 1 medicina-59-01780-f001:**
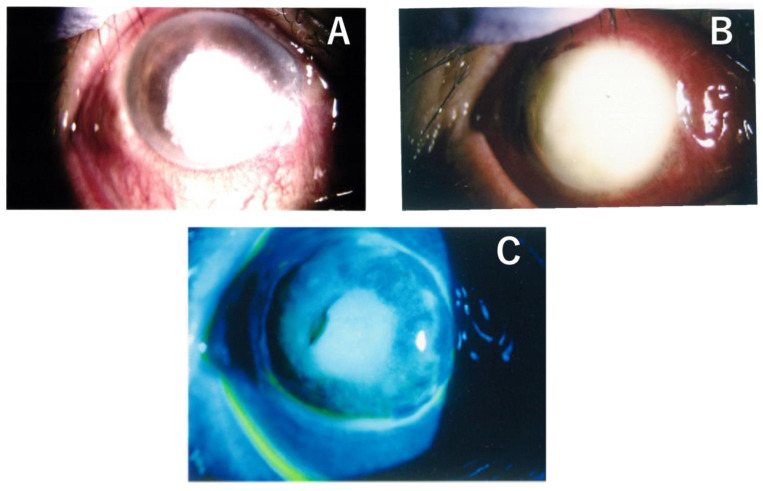
The right cornea showed a large centrally infected ulcer with grayish-whitish infiltration involving the central cornea and extending to the lower periphery (**A**). The corneal ulcer and infiltration after two weeks of the first visit (**B**). The right whole cornea descemetocele after three weeks of the first visit (**C**).

**Figure 2 medicina-59-01780-f002:**
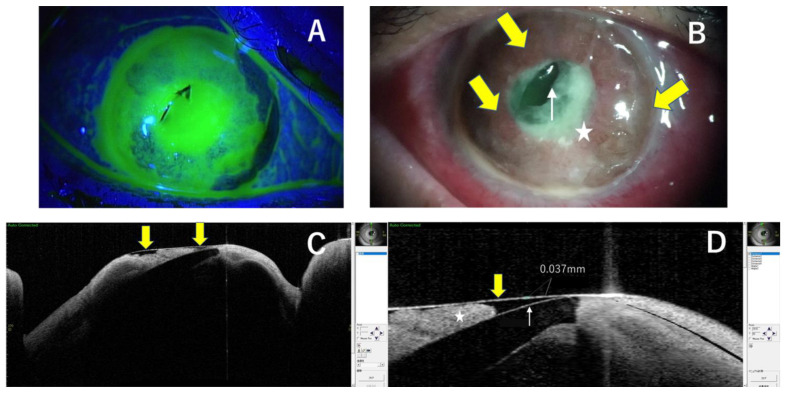
The right corneal ulcer stained greenish with fluorescein, involving nearly the whole cornea, which was almost completely melted down with the remaining DM (**A**). Descemetocele (yellow arrows) protruding forward, the pupil area was filled with melted necrotic material (star), and the intraocular lens was partially protruding from the pupil and indenting the DM (white arrow) (**B**). The vertical corneal optical coherence tomography (OCT) examination revealed 37 µm of corneal thickness attached to its back surface with the iris (star) and part of the IOL (white arrow) was protruding through the pupil and indenting the DM (yellow arrows) (**C**,**D**).

**Figure 3 medicina-59-01780-f003:**
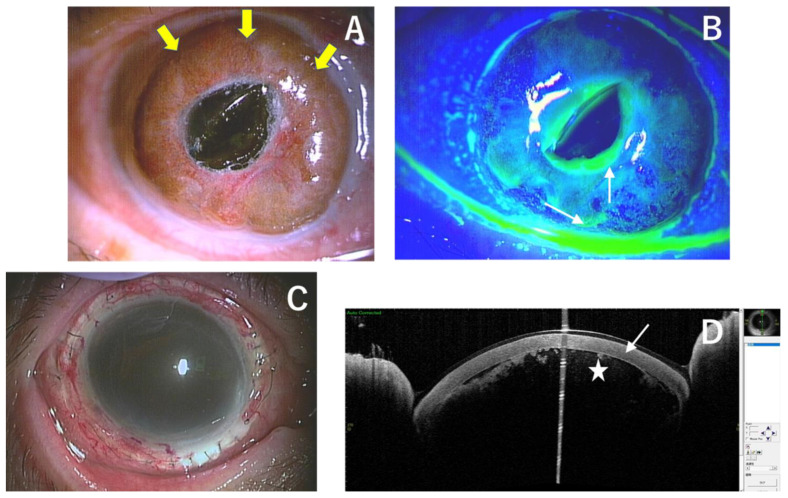
Right corneal descemetocele (yellow arrows) showed resolution of ulceration and infiltration except for a few scattered greenish spots stained with fluorescein (white arrows) (**A**,**B**). Right eye after corneoscleral transplantation (**C**). Vertical OCT of the right eye after corneoscleral transplantation (white arrow) with the formation of the anterior chamber (star) (**D**).

## Data Availability

The datasets used during the current study are available from the corresponding authors upon reasonable request.

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
