# Peer review of "Whole Corneal Descemetocele"

_medicina, 2023, doi:10.3390/medicina59101780_

Round 1

Reviewer 1 Report

The authors present a case thought to be a whole corneal descemetocele.

1.       Anterior segment photographs are of poor quality.

2.       Lines 51-54: These sentences should be moved to the Discussion section.

3.       In the Figures, explanative arrows are missing nearly everywhere. This complicates the reading for the non-ophthalmology-educated readers. Please add arrows.

4.       The inset should be added to depict the direction and location of the scan in OCT images.

5.       The presence of a whole corneal desmatocele is not evident in OCT images.

6.       The term cystocele is awkward. It should probably be abandoned. The whole corneal descemetocele can be satisfactorily used.

Author Response

Response to Reviewers

First Reviewer

Summary:

2. Questions for General Evaluation

Reviewer’s Evaluation

Response and Revisions

Does the introduction provide sufficient background and include all relevant references?

Yes/Can be improved/Must be improved/Not applicable

We appreciate your comment

Are all the cited references relevant to the research?

Yes/Can be improved/Must be improved/Not applicable

We added some references to improve the manuscript

Is the research design appropriate?

Yes/Can be improved/Must be improved/Not applicable

We added some sentences to improve it

Are the methods adequately described?

Yes/Can be improved/Must be improved/Not applicable

-

Are the results clearly presented?

Yes/Can be improved/Must be improved/Not applicable

We added arrows and stars to clarify the results

Are the conclusions supported by the results?

Yes/Can be improved/Must be improved/Not applicable

We tried to the improve it

Comments and Suggestions for Authors

The authors present a case thought to be a whole corneal descemetocele.

Response: We are grateful for the review and valuable comments. We have addressed each of your comments and added point-by-point responses below.

  • Anterior segment photographs are of poor quality.

Response: We are grateful for your insightful correction. We changed the photograph to a better quality but unfortunately these photos were sent to us from the primary ophthalmic clinic which examined the patient.

  • Lines 51-54: These sentences should be moved to the Discussion section.

Response: We are grateful for your suggestion. We moved these sentences to the second paragraph in Discussion section as per your suggestion.

  • In the Figures, explanative arrows are missing nearly everywhere. This complicates the reading for the non-ophthalmology-educated readers. Please add arrows.

   Response: We are grateful for this suggestion. We added arrows and stars to make it easier for understanding as per your suggestion.

  • The inset should be added to depict the direction and location of the scan in OCT images.

Response: We are grateful for your suggestion. We added “Vertical” to explain the direction of the of the OCT images as per your suggestion. Also, we changed the photos to better resoluion photos to clarify the inset which is present in the right upper corner of the photo revealing the direction and location of the scan.

  • The presence of a whole corneal descemetocele is not evident in OCT images.

Response: We thank you for your comment. We explained that the whole iris is bowed anteriorly and attached to the back surface of DM. So, it is masquerade by presence of iris except in the pupillary area where iris is not present. In the pupillary area, we could estimate the thickness of DM easily as shown in Figure 2 C, D.

  • The term cystocele is awkward. It should probably be abandoned. The whole corneal descemetocele can be satisfactorily used

Response: Thank you very much for your suggestion. We deleted this term from the whole manuscript as per your suggestion.

Reviewer 2 Report

The article is quite interesting, anyway i'm not able to read if you done a swab for bacterial and fungal infection in order to prescribe a correct local and systemic therapy, if was not done, You should be specify the reason ( pain, patient compliance ....) 

On the other hand we can define Whole Corneal Descemetocele (Corneocele) like a total corneal melting that required a limbus keratoplasty, I suggest pathological definition review.  

The quality of English could be improved in syntax adjustment. 

Author Response

Response to the Reviewers

Second Reviewer

Summary:

2. Questions for General Evaluation

Reviewer’s Evaluation

Response and Revisions

Does the introduction provide sufficient background and include all relevant references?

Yes/Can be improved/Must be improved/Not applicable

We appreciate your opinion

Are all the cited references relevant to the research?

Yes/Can be improved/Must be improved/Not applicable

We appreciate your opinion

Is the research design appropriate?

Yes/Can be improved/Must be improved/Not applicable

We appreciate your opinion

Are the methods adequately described?

Yes/Can be improved/Must be improved/Not applicable

We appreciate your opinion

Are the results clearly presented?

Yes/Can be improved/Must be improved/Not applicable

We added arrows and stars to clarify the results

Are the conclusions supported by the results?

Yes/Can be improved/Must be improved/Not applicable

We tried to the improve it

Comments and Suggestions for Authors

The article is quite interesting, anyway i'm not able to read if you done a swab for bacterial and fungal infection in order to prescribe a correct local and systemic therapy, if was not done, You should be specify the reason ( pain, patient compliance ....)

Response: We are grateful for the review and valuable comments. We have addressed each of your comments and added point-by-point responses below.

  • The article is quite interesting, anyway i'm not able to read if you done a swab for bacterial and fungal infection in order to prescribe a correct local and systemic therapy, if was not done, You should be specify the reason ( pain, patient compliance ....)

Response: We are grateful for your insightful correction. We added the sentence “Conjunctival swab was taken from the patient for culture and sensitivity in a trial to identify the causative organism.” To clarify that we did it in our procedure of diagnosing the case as per your suggestion. Also, we already mentioned that “Smears and cultures performed at the time of initial examination were negative for bacteria and/or fungi.” At the end of case presentation and before Discussion section.

  • On the other hand we can define Whole Corneal Descemetocele (Corneocele) like a total corneal melting that required a limbus keratoplasty, I suggest pathological definition review.

Response: We are grateful for this suggestion. We deleted the term (Corneacele) from the whole manuscript and define it as “Whole corneal descemetocele”. In the Discussion section, we explained that “In the present study, our only options were to control and suppress the inflammation along with the use of PRP as a corneal graft, due to the size of the descemetocele that involved the whole cornea.”

  • Comments on the Quality of English Language

The quality of English could be improved in syntax adjustment.

Response: Our manuscript has been edited by a native English speaker at Forte, Inc. Forte is a Japan based company that employs editors and rewriters with a science background. That company has been helping the Japanese scientific community to publish in international journals since 1987. We will attach this certificate to be examined by the Editor and Reviewers.

Round 2

Reviewer 1 Report

Necessary changes have been made by the authors.

Reviewer 2 Report

Now the article could be published, but I'm not able to read the microbiological result of swab, If the result is negative, please specify.